# "Just Be Careful, Since Social Media Is Really Not as Safe as It's Being Portrayed": Adolescent Views on Adult Support for Safer Social Media Use

Miroslava Tokovska [1,*], Ragnhild Eg [2], Ashley Rebecca Holt Bell [2] and Merete Kolberg Tennfjord [1]

1  Department of Health and Training, Kristiania University College, 0152 Oslo, Norway
2  Department of Psychology, Pedagogy and Law, Kristiania University College, 0152 Oslo, Norway
*  Correspondence: miroslava.tokovska@kristiania.no

**Abstract:** Social media is an integral part of the lives of adolescents, but they are also closed arenas concealed from the next of kin and are resistant to parental mediation. Consequently, this study aims to investigate how adolescents reflect on the safe use of social media and the conversations they have with their parents. The present study used data from 8 focus group interviews with adolescents aged from 15 to 19 years in Norway. Data were analysed using systematic text condensation—an inductive thematic cross-case analysis. The results showed three themes: (1) next of kin encourage caution, (2) safety is created through mutual learning, and (3) adolescents expect parental mediation. The present study has important implications for policymakers, media educators, and education institutions and its findings will enable better preparation and targeting of curricula and education from basic to secondary schools on a national level.

**Keywords:** adolescents; parental mediation; next of kin; social media; Norway; a qualitative study

## 1. Introduction

Social media has become an integral part of life for Norwegian adolescents. By the age of 13 nearly every teen has at least one social media profile, and the young generally spend hours online every day; social media take up a lot of this time (Norwegian Media Authority 2020; Statistics Norway 2021). At the same time, parents and other carers (the next of kin) remain the main socialising agents and gatekeepers for the younger generation in terms of accessing media and digital devices (Iqbal et al. 2021a). Available research indicates that parents play an important role in shaping their children's tolerance. The mediation of online content by the parent is a social process guided by the parent and child's differing social norms, which enriches the adolescent's overall socialisation processes (Daneels and Vanwynsberghe 2017; Pfaff-Rüdiger and Riesmeyer 2016). Although most children who are online view it as a positive experience, many parents, teachers. and other carers worry that their children's engagement with screens comes with negative consequences, including detriments to mental health such as depression, anxiety, or even internet dependency (Kardefelt-Winther et al. 2022; Singh 2022).

Social media has evolved since its early days, from forum and photo sharing formats to personally tailored services. The platforms now register a user's every action, analyse the acquired data, and then filter and prioritize the outputs deemed most relevant to the user (Powers 2017; Swart 2021). Despite the widespread use of these social media platforms, parents and other carers seem to have little insight into these unique online worlds and how young people navigate the changing digital landscapes (Sarwatay et al. 2021). Even more alarming, 55% of Norwegian parents lack knowledge of the age limit for social media platforms such as Snapchat, Instagram, YouTube, TikTok (Norwegian Media Authority 2020), which may indicate that parents do not understand the digital world that young people live in. Furthermore, the impact of personalized social media content on potentially

susceptible young people in connection with parental mediation has received insufficient research attention.

Adolescence is defined by development and changes, making social media a part of the cognitive, emotional, and social maturation that occurs early in life. This is a cause of concern among the Norwegian authorities (The Ministry of Health and Care Services 2019; Norwegian Media Authority 2020), perhaps amplified by findings that adolescents may not be as digitally competent as they believe themselves to be (Porat et al. 2018; Bell et al. 2021). Digital competence covers a range of abilities and proficiencies, including digital literacy, social media literacy, and algorithmic literacy, but it also covers attitudes, cooperation, and self-regulation, to name a few (Khan and Vuopala 2019; Porat et al. 2018). Whereas digital literacy covers both the technical and cognitive skills needed to use and evaluate online platforms, social media literacy is specific to the social platforms (Manca et al. 2021; Porat et al. 2018). In short, digital competence is multi-faceted and includes many aspects of human interactions with online technology, whereas the literacy terms allude to skill sets for specific contexts.

Supportive parent mediation with adolescents holds the most promise for enabling them to draw maximum benefit and minimum risk from online life (Livingstone et al. 2017; Nichols and Selim 2022). Despite a few contradictory findings, the majority of the work teaching adolescents suggests that parents, teachers, and other next of kin play a role in promoting safe social media habits among adolescents (Livingstone et al. 2021). According to Clark (2011), parental internet mediation acknowledges that parents actively manage and regulate their children's internet use, while mitigating its negative effects on teens. Hence, five dimensions of parental internet mediation were developed, keeping in view the specific attributes of the internet. These are: (1) active co-use or instructive mediation, where parents encourage, share, and discuss internet use mutually; (2) active mediation of internet safety, where parents guide teens towards safer online practices; (3) restrictive mediation, where parents set rules and regulations; (4) monitoring, where parents check the record available afterwards; and (5) technical mediation, where parents use software or control mechanisms to restrict, filter, or monitor online activities (Livingstone et al. 2015). Several other studies emphasise the importance of mediation by next of kin in developing digital literacy and coping with online risks (Cabello-Hutt et al. 2018; Livingstone et al. 2017; Santos et al. 2019), but not without exceptions (Purnama et al. 2021). It is worth remarking that not all parents have the necessary digital competence themselves (Tomczyk 2018). Others have noted that a specific approach to mediation may make a difference. For instance, one study found that restrictive mediation by parents did diminish their children's online risks, but it did so by restricting their opportunities to learn from online encounters (Rodríguez-de-Dios et al. 2018). Researchers found no such correlation when considering active parental mediation (PM); instead, the authors point out that the more time adolescents spend online, the more digitally literate they become, and the more benefits they are able to sample from the technology (Rodríguez-de-Dios et al. 2018). The benefits of the use of social media can be a source of emotional support, connection with like-minded peers, getting help online, and less loneliness (Ghai et al. 2022). This stresses the importance of pursuing research on who and what contributes to adolescents' awareness, knowledge, and skills surrounding social media and online safety (Hurwitz and Schmitt 2020; Livingstone et al. 2021; Manca et al. 2021). Moreover, increasing adolescents' digital safety is a relevant concern for parents, educators, and policymakers.

The theory of PM is based on a resilient approach and has evolved to consider the interactions between parents and children concerning the use of social media to mitigate the effects of communication in the digital world. This study used PM as the theoretical framework, and explored the following research question: *How do adolescents reflect on their conversations with parents, and how do they use social media in a safe and sensible manner?*

## 2. Literature Review

Adolescents are very active on social media (Bakken 2021), which implies that they are exposed to social influence both in their physical and digital lives. Researchers, teachers, next of kin, and government agencies are concerned about the potentially harmful influences of social media; these concerns relate to the ongoing increases in the time that adolescents spend in front of screens (Elhai et al. 2021), the negative and even dangerous communications they may encounter online (Hjetland et al. 2021), as well as the perceived vulnerability of young people (Valkenburg et al. 2022). According to Pons-Salvador et al. (2022), most parents are aware of the benefits and risks of the Internet, but many of them did not know what to do to protect their children. More than half of the parents said they did not know how to set up content filters or manage parental control, and 40% did not know how to block unwanted advertisements.

In 2016, the rapidly evolving nature of social media accelerated (Isaac and Ember 2016; Haynes 2016; Newton 2016). Since the introduction of algorithmic feeds, researchers from different disciplines have taken an interest in the impact of personalization algorithms, for instance on opinion formation, body image, and practical use (Harriger et al. 2022; Wiard et al. 2022; Cotter 2022), but there is very limited knowledge on young adolescents' interactions with algorithmic outcomes. Awareness, knowledge, and understanding of the functions and purposes of algorithms can be challenging for both teenagers and parents, even though they deal with them daily on social media. Moreover, the digital competence required to navigate social media safely requires users to keep up with rapidly evolving technology. It also requires deep knowledge and understanding of how data-driven and automated services influence our behaviour, decisions, and social processes.

This study was part of a larger project focusing on how adolescents experience personalized content on social media. During group interviews, we registered how teenagers think about the safe use of social media and how they converse with their parents on the matter. According to Head et al. (2020), we need to consider steps for educators, next of kin, and government agencies to better prepare and protect adolescents for an expanding digital world and to understand the technological and social forces that shape the flow of news and information. This study provides early insight on the skills and knowledge that the young require to competently navigate personalized social media.

## 3. Materials and Methods

The current study used data from eight focus group interviews performed between April and June 2021; social media habits, including online safety, were assessed. Focus group interviews represent participants' varied experiences and support associations, variations, and different aspects of experience on a selected topic (Plummer 2017), and were thus considered an appropriate research method for the topic in question. This qualitative study presents one part of a larger qualitative project that addresses the social media habits among adolescents aged 15 to 19 years in Norway. From the overall project, themes related to awareness, familiarity, and emotions toward personalised content on social media have been published (Bell et al. 2021).

### 3.1. Sample

The adolescents were recruited from one public lower secondary school and one public higher secondary school in an urban area in South-East Norway. Recruitment was done in randomly chosen classes by the teacher responsible for each class. The adolescents received written information about the study, including a consent form to sign. Written informed consent forms were also sent to the parents of the youngest participants (those aged 15 years). In addition, all adolescents were encouraged to inform their parents about the project. If the number of adolescents interested exceeded the number needed for the project, a random draw was performed, and a waiting list was compiled of those who were not chosen.

The original sample included 48 participants (21 male and 27 female students). However, 4 withdrew from the study before the interviews began and were replaced by participants on the waiting list. The reason for withdrawal was due to illness (n = 3) while the fourth did not give a reason. The youngest participants (n = 24) were recruited from the last year of lower secondary school (aged 15 or 16). These were assigned to four focus group interviews: two groups for males and two groups for females. The other participants (n = 24) were recruited from three upper secondary school levels (aged 16–17, 17–18, and 18–19). One male focus group and one female focus group were assigned for the age group 16–17 years, while one mixed-gender group was assigned for the age groups 17–18 and 18–19 years. In the mixed-gender group, one male and five females participated. One male participant was excluded from the analysis due to being older than the other participants, and therefore considered not being representative of the age-group studied. Thus, the final sample consisted of 47 adolescents: 20 males and 27 females to be analysed.

### 3.2. Data Collection and Materials

We used an interview guide with open-ended questions to ensure a coherent narrative of experiences with online safety, with focus on parental mediation. The interview guide was tested beforehand on a panel of four adolescents between the ages of 13 to 15 years. Each focus group interview lasted between 60 to 90 min, including time for explaining the study purpose, collecting the signed consent forms. and debriefing. From the larger study, we obtained most of our information by analysing the question that was directly related to adolescents' experiences with online safety: "Have you discussed with your next of kin, parents or friends how you can use social media in a safe and sensible matter?" However, since this topic was discussed throughout the interviews, we acquired additional data from replies to other questions about the use and knowledge of social media. Two researchers were present during all interviews. Only one interviewer (ARB), trained in qualitative interview techniques, led all the interviews, and the second researcher (MKT or RE) took notes on non-verbal communication. The interviews were held in a separate room at the respective schools, and all members of the focus groups were seated in a circle allowing everybody to see each other. The interviews were recorded on a secure online service for recording and storing research data (Nettskjema—Norwegian Net-Based Tool for Data Collection n.d.).

The participants in each focus group were familiar with each other due to the focus groups including adolescents from the same class or the same class level. The participants and the researcher had no prior relationship.

### 3.3. Data Analysis

The transcriptions were analysed using systematic text condensation, which is an inductive thematic cross-case analysis (Malterud 2012). The analysis was conducted in four steps, as elaborated in Figure 1.

### 3.4. Ethics Statement

The Consolidated Criteria for Reporting Qualitative Research (COREQ) (Tong et al. 2007) were followed. The study and its data handling were approved by the Norwegian Centre for Research Data (Ref. Nr. 644850). All participants were given a gift card for their time, valued at around EUR 30. Adolescents were informed both verbally and in writing about the purpose of the study, and their ethical and voluntary rights. Informed consent forms were given to the participants through their teacher several days before the interview took place. Hence, adolescents were able to read the informed consent forms alone prior to the interview and decide if they wanted to participate. Parents of adolescents under the age of 16, those aged 15, were also given informed consent. Signed consent forms were given to the research team at the time of the interview. Anonymisation was ensured by not using names during the interview, and by giving each student an alias when transcribing the interviews. Although the participants were informed that they could request their

statements for comments and corrections, none did so. The present study used fictive names in the results.

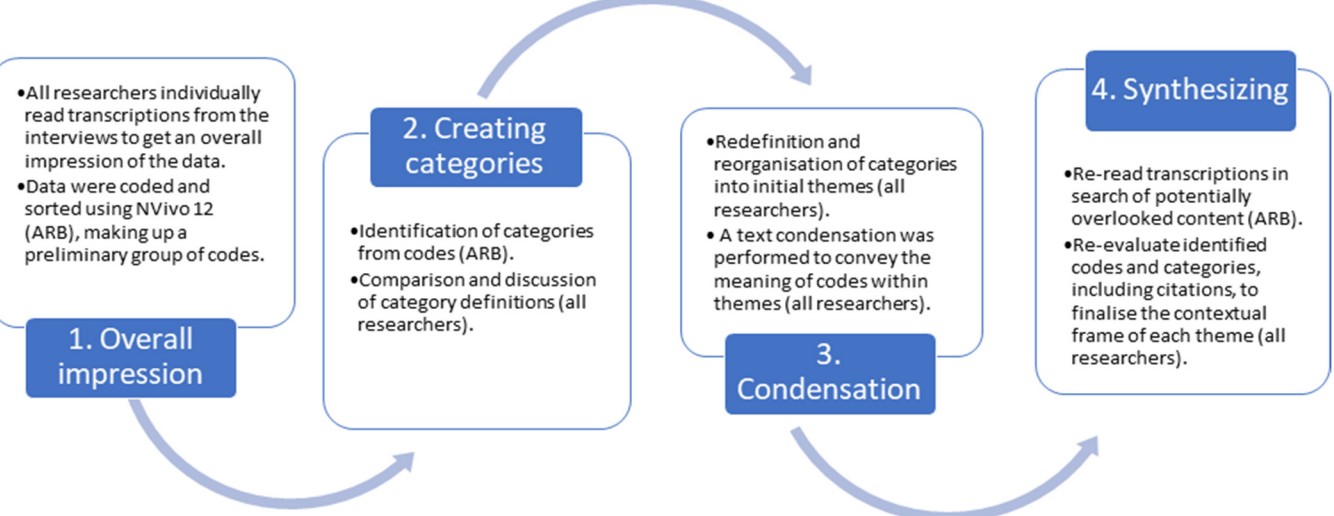

**Figure 1.** The analytical process uses systematic text condensation (Malterud 2012).

## 4. Results

The current study investigates how adolescents reflect on their conversations with adults, primarily their parents, regarding their use of social media in a safe and sensible manner. Through systematic text condensation we identified three themes: (1) next of kin encourage caution, (2) safety is created through mutual learning, and (3) adolescents expect parental mediation. In the following text, illustrative statements will be presented to ensure a transparent interpretation of the themes. Statements have been translated from Norwegian to English.

### 4.1. Next of Kin Encourage Caution

The conversation with parents about how to use social media was prominent in our interviews among adolescents. Hannah (15–16 years old) said: "When I was allowed to have social media, I remember that my mom went through everything with me. Like, 'post this' and 'do not post this'." Jacob (15–16 years old) from a different interview shared having had similar conversations as Hannah, with his parents and other adults, while expressing agreement with what the adults had told him; "Just be careful, since social media is really not as safe as it's being portrayed." Throughout the interviews, most participants expressed that their parents had wanted them to be more aware, and possibly try to help them create an understanding of the various risks that may occur when using social media. "Mom and dad have talked a lot to me about the fact that I must be careful who I talk to, what I post and what I say on social media, because it can be misinterpreted. Suddenly you might be talking to a 40-year-old person, who you thought was really 15 years old", said Abigail (15–16 years old).

Although it seemed like the adolescents found it reasonable to have conversations with next of kin about the risks that may arise while using social media, some expressed no need to talk with parents or other adults, at least not anymore. For example, Hannah, who previously had shared her social media-safety conversation with her mother said: "But now you understand stuff like that yourself", referring to it being okay to have conversations with a parent about social media safety during social media debut, but it was not needed later. Among the older males from a different interview, Andrew (17–18 years old) added: "I personally don't talk as much with my parents about it [safe use of social media]. Also, I feel like I have a pretty good understanding of how it works and how I should sort of behave online."

The school may also play an important role in teaching adolescents how to safely use social media. A majority of the adolescents mentioned that social media and internet safety had been a topic at school. For example, Andrew said: "We have had those 'safety on social media' courses, both during primary and secondary school." It is not only the teachers who raise adolescents' awareness of social media safety—the school nurses also play an important role. "We've had school nurses doing presentations about it [social media and online safety], there was a lot of it during lower secondary school ( . . . ) but I don't think there are many people [students] who actually care" said Sophia (16–17 years old), with many of her fellow classmates expressing agreement. Emma (16–17 years old) elaborated shortly after: "My mom has always said 'do not chat to strangers on your phone', um and that might have happened, but I'm still fine", she reported while laughing softly.

To summarise, adolescents' next of kin encourage caution toward adolescents' social media usage. Having conversations with parents about social media safety before a social media debut seems to be appreciated by adolescents, especially when they get older. Nevertheless, some adolescents do not necessarily listen to school nurses or parents, and the school nurses' involvement might not be as helpful as intended. Thus, our findings should be interpreted with caution.

### 4.2. Safety Is Created through Mutual Learning

Based on our interviews, it seems that adolescents find it important to learn how to use social media in a safe and sensible way. Adolescents reflected on the importance of mutual help and learning in their own families, and also among groups of friends and fellow classmates. However, throughout the interviews it became clear that many of the participants had an impression of their parents as being digitally outdated, which again can be a reflection of being digitally incompetent. "They are completely outdated", said Sophia with an exasperated tone in her voice, followed by a light laugh which was met by her fellow classmates. A majority of the participants stated that even though their parents used social media actively, the platforms differed from their own; parents used mainly Facebook and Snapchat, while the adolescents themselves used Snapchat, TikTok, and Instagram. In other words, the parents were behind the times when it came to social media trends. Still, a majority of the participants across all interviews noted that they usually followed, and were followed by, their parents on some of the social media platforms—Snapchat in particular. Some participants even pointed out that their parents were familiar with the various functions of social media. Isabella (15–16-years old) illustrated this by saying: "You notice that parents [these days] have become a little more modernised. Nowadays it's, for example, their focus on the use of Snap Map [a function on Snapchat] so they can see your position." Several of the youngest participants confirmed the fact that parents had told and reminded them to use such functions, in order for the parents to have a good overview of their child's whereabouts.

Even though the adolescents shared mixed opinions of their parents' digital competence, almost all participants said they had given parents or other adults technical support. This was expressed in several ways. For example, Justin (16–17 years old) said he had helped his mom: "I taught my mother Snapchat. She discovered not too long ago how to save photos taken on Snapchat [thanks to my help]." In many ways, adolescents had their own important roles in the family, supporting family members and showing them the functions of social media applications.

A few also explained that they had given social media safety advice to siblings or younger friends even though they had not received advice themselves, mainly due to their parents' lack of knowledge in this fast-growing digital world. Mason (16–17 years old) said for example: "I haven't received as much advice myself since it [social media] was pretty new when I got it. But I do give my little sister advice about what she should be mindful of, and what she shouldn't do [while using social media]. I've also helped my grandmother now that she has started using Snapchat." In general, the advice shared across most of the interviews was mainly focussed on not talking or chatting to strangers online. For example,

Mason replied the following when asked what kind of advice he gave to younger peers: "Only have friends on Snapchat in the beginning, at least until you're older and capable of taking care of yourself, like we are now. But when you're 13, 14, 15 years old, only have your friends on Snapchat." Several participants across different interviews agreed with Masons' statement. In addition, one participant even mentioned that he had unknowingly had a male paedophile as one of his Snapchat contacts, without ever talking to him; he was informed by the police. After this incident, he became cautious about accepting friend requests from strangers and has also warned others on how to take safety measures when using social media.

Although teenagers criticise the digital competence of their parents and close relatives, they pass on their experiences and understand that they are a new generation that learns by doing. The advice the adolescents received from parents, but also the advice they themselves gave to younger peers, was mainly focussed on not adding strangers on social media. This mutual sharing of advice can be understood as creating safety through mutual learning in connection with the use of social media.

### 4.3. Adolescents Expect Parental Mediation

On the one hand, adolescents expressed an expectation that parents would be mediating and advising them in regard to their social media use. The youngest participants from lower secondary school focussed mainly on how they liked, or would like, parents to be aware of their children's safety on social media. As Ethan (15–16 years old) said: "It's easy to get lost in a platform, and perhaps it's not that safe. So, it's good to know what you should and shouldn't do." Older adolescents in upper secondary school focussed mainly on how they would like parents in general to be more aware of the safety issues on social media when dealing with younger adolescents. In other words, they felt capable of using social media safely themselves. Andrew elaborated: "It's important that parents engage in their child's use of social media, especially during primary school and perhaps lower secondary school as well ( . . . ). When we get older, we can use and handle it [social media] better." It seems, however, that parents should not just focus on and discuss the negative aspects of social media, but they also should make room for conversations about social media's positive aspects. Ava (18–19 years old) was one of the participants who indirectly expressed a wish for more nuanced conversations about social media: "I believe we talk more about what you should not do, rather than what's smart do to [with adults/parents] ( . . . ) It's often expected [by adults/parents] that you behave when online."

On the other hand, the pattern seems to be that parents do not keep up with their children's social media. As Sydney (16–17 years old) said: "I believe that many parents only have social media because, as they say: 'we are looking after our children and checking who they are following [on social media]'. But in reality, they don't understand [social media], so it doesn't really help." Should parents be more present, and take better action to understand their children's digital world? According to some of our participants, the answer is yes. In an interview with the youngest girls, parental involvement and lack of understanding was a topic that the participants themselves raised. Chloe (15–16 years old) summed up their expectations from their parents: "I sometimes feel that they [parents] don't try hard enough [to understand social media]. We are their children, social media affects our mental health, our views of ourselves, and how we develop as a person. And it's the parents' job to make sure we are okay ( . . . ). So when it comes to social media and the massive pressure on the platforms, I think it's stupid of parents to say, 'just don't care or think about it'. It's not that easy, it's more complicated than that". Chloe, along with fellow classmates, expressed a wish of having more understanding parents, who perhaps tried a little harder to comprehend their child's digital world. However, as Elizabeth (15–16 years old) said: "I don't think they [parents] will ever understand one hundred percent since they haven't grown up with social media ( . . . ). So then you'd rather talk with your friends if there's something you're struggling with online." Does parents' lack of keeping track of their child's digital world affect the bond between child and parent? Perhaps it is too late for the

current generation of parents, but it is possible that future generations will be less afraid to follow the latest digital trends, and in that way they will be more present for their children and teach them how to safely use social media. As Ryan (16–17 years old) said: "I feel that our generation are the ones who have to, in a way, figure it [social media] out ourselves." Adolescents have expectations from their next of kin regarding digital competencies, but they are also aware they must "learn by doing" because their next of kin do not have enough knowledge to help them. The adolescents' peers are the ones who, rather than the parents, encourage learning and share experiences for safe social media use.

## 5. Discussion

This study aimed to investigate how adolescents reflect on the safe use of social media and on their conversations with parents. Our findings show that parental mediation does take place in the family, but that both the school and peers also contribute to the learning of safe social media use. Next of kin mediation in teenagers' social media use is a multifaceted concept used to regulate teens' online activities and enhance a resilient approach to reduce the risks associated with the use of social media. The concept of parental mediation originated primarily from media studies, especially in the areas of television and video games, on the effects of media content on young people's behaviours (Rothfuss-Buerkel and Buerkel 2009). Clark (2011) and Livingstone et al. (2015) described active parental mediation, which mitigates the negative effects on teens. The results of our study show that parents use four out of five dimensions of parental internet mediation, such as: active co-use or instructive mediation, where parents encourage, share, and discuss mutually; active mediation of internet safety, where parents guide teens towards safer online practices; restrictive mediation, where parents set rules and regulations; and monitoring, where parents check the record available afterwards, e.g., the use of Snap Map. Regarding technical mediation, where parents use software or control mechanisms to restrict, filter or monitor online activities, our adolescents did not mention it. In our study, we did not identify the dimensions of parental internet mediation. Based on the interviews, adolescents emphasised that awareness and information are needed for the feeling of safety and security, where information is passed on by peers and next of kin. The Norwegian Media Authority (2020) stressed that 69% of Norwegian 9–18-year-olds reported that they were under the age of 13 when they became a user of Snapchat, Instagram, TikTok, and/or YouTube. This is despite the age limit of social media platforms, which is 13 years of age (Personal Data Act, Section 5). The need for parental mediation is all the more pronounced the younger the age of the child when they first use social media. This is indirectly confirmed by the adolescents in our study, who emphasised the need for caution and the need to feel safe.

Spies Shapiro and Margolin (2014) stressed that learning about social media safety at school might be more helpful than they themselves realise, considering that the students are at a formative age where they are suspectable to social influence. Notten and Nikken (2016) described that parental involvement has great impact on teens' online activities on social media; additionally, parents' digital skills, as well as active mediation by co-using the internet, were important factors in reducing risk-taking, especially among adolescent boys. The family is still the first and primary place for children's contact, and positive family interactions contribute to the development of social responsibility among adolescents (Cheng et al. 2021; Nathanson 2015). Parents can also monitor adolescents' friends on social media and inquire about names they are not familiar with. This is extremely important to protect adolescents from potential cyberstalkers and cyberbullying. The authors stressed that it is essential to educate adolescents, as well as their teachers and parents, on cyberbullying, digital identity, the impact of digital footprints, and the use of inappropriate social media (Martin et al. 2018). Parents should be present and offer practical advice and warnings when adolescents start using social and digital media so that challenges related to using media in a safe and sensible way are ensured. The school can also make a significant contribution, and teachers should repeatedly talk about online rules in various teaching

sessions. It is important that there is constant work needed to design good preventive measures aimed at young people. Our participants expressed the opinion that the school (teachers as well as school nurses) is successful in actively paying attention to the safe use of social media. De Leyn et al. (2021) show that the school environment was considered an appropriate space for "becoming digital media literate". As more teaching takes place online, it is important that adolescents receive training on how to use social media safely through schools and school staff (including school nurses). These skills need to be repeated every year due to the rapid development of algorithms, new applications, communication platforms, and various digital devices.

One way to achieve online safety through mutual learning could be to develop intergenerational intervention programmes or peer intervention programmes, where parents and adolescents or peer adolescent groups learn together at first with the help of parents or next of kin (Rubach and Bonanati 2021). These programmes could be formed and modified in connection with the development of safety in the virtual world and the digital competencies of adolescents, and they would likely foster specific forms of parental involvement, such as a particular interaction between parents and adolescents. Another option could be to offer parents the opportunity to reflect on their interactions with their child. The process of reflection allows parents to mentor children in a correct way and support them while they learn to use social media in a safe way. The study by Iqbal et al. (2021b) notes that, based on parents' perceptions, it is important to support the suggestion of launching government-supported initiatives and updating the curriculum module. This would ensure that parents, teachers, professionals, and communities are aware of potential online risks, online protection tools, and safer internet best practices so that a safe online environment is created for the younger generation. It is necessary to note that each parent's conversation or discussion with adolescents can help the adolescent develop more decision-making skills and become aware of the safe use of social media.

Adolescents' social trust may be shaped by positive family experiences, which is consistent with the notion that the evaluation of trustworthiness is based on one's social experiences (Draude et al. 2018). According to a research report by Letnes et al. (2021), parents' perception of risk was, among other things, connected to whether strangers had the opportunity to contact their children, privacy and security, and possible exposure to harmful content. Parents, according to Letnes et al. (2021), are concerned that their children may post personal information or inappropriate images online, or that others may exploit photos of their children. Our adolescents in the present study reported that through applications such as the Snap Map parents could maintain control over their location. Even though our findings indicate that such applications have been found to be popular amongst parents, having trust in their children and conversations about online safety are important and have significant benefits.

Despite the importance and relevance of the topic under investigation, this study has some strengths and limitations. We consider the qualitative approach using focus group discussions as the first strength of this study. Focus groups are a naturalistic (i.e., close to the everyday conversation) approach and include dynamic group interactions, which provide in-depth insights and encourage participants to explore and clarify perspectives (Tong et al. 2007). Secondly, in order to increase internal validity, the interviews were all led by the same researchers; additionally, two researchers with experience in qualitative research supervised the data analysis. Thirdly, both mixed and same-sex focus groups were used to enable a variety of opinions and interactions during the discussions. There are also some limitations to this study that are important to mention. Although the purpose of the study was explorative in nature, it is possible that determinants have been missed due to the nature of the research—themes, factors, and effects were identified and explored as the data presented itself. The sampling procedure was administered by the school itself in order to preserve the anonymity of participants. Although the administrators were instructed to do random selections, it could have biased the sampling. Furthermore, as we have no data on socio-economical background factors our sample may not be representative of the general

adolescent population. Importantly, since the study was conducted in Norway, where the internet and social media are widely used (Bakken 2021), future research should examine population subgroups within Scandinavian countries where social media is less common and investigate rating scales to gauge diverse dimensions of parental digital literacy in relation to the safe use of social media. The future inclusion of more countries in the research would also be beneficial for understanding parental mediation across cultural contexts.

## 6. Conclusions

The goal of this study was to investigate how adolescents reflect on the safe use of social media and their conversations with parents. The results suggest that parents are likely to instruct adolescents on the use of social media, but the instructions were often in the form of restrictions. In the same way, peers and school employees are also involved in the safe use of social media. The mutual passing of good advice regarding the use of social media was shown to happen spontaneously in families as part of generational interactions, and also between peer groups. Since many adolescents perceive their parents and teachers as lacking in knowledge about social media, we face a challenge in raising their level of competence. One step towards this challenge is to better specify and continuously update the teaching curriculum, ensuring that it follows technological developments. On the one hand, young people need safe frameworks, understanding, and support when using social media. On the other hand, teachers already struggle with time constraints and keeping their material up to speed. Consequently, the responsibility of updating the curriculum needs to be allocated to decision makers that also have the capacity to follow up the practical teaching. Furthermore, mutual reliance can be an aid. Adults rely on adolescents to safely use social media and adolescents rely on adults to help them in case of problems. This mutual reliance improves interactions, connections, and feelings between immediate relatives or peers. In this way, personal awareness can be raised, in turn facilitating the spread of knowledge on safe social media use.

The present study has important implications for policymakers, media educators, and educational institutions and its findings will enable better preparation and targeting of curricula and education from basic to secondary schools on a national level. Given that parents and schools are the primary caregivers and socialisation agents for children, it is important to understand what parents and schools can to do increase the safe use of social media and maximise opportunities brought about in the digital age. The parental mediation of children's safe use of social media cannot be taken for granted. Next of kin need similar support and an increase in digital literacy in connection with the safe use of social media.

**Author Contributions:** Conceptualization, R.E. and M.K.T.; data curation, M.T. and A.R.H.B.; formal analysis, A.R.H.B.; funding acquisition, M.K.T.; investigation, R.E., A.R.H.B. and M.K.T.; methodology, M.T., A.R.H.B. and M.K.T.; project administration, M.K.T.; resources, R.E. and M.K.T.; software, A.R.H.B.; validation, M.T., R.E., A.R.H.B. and M.K.T.; visualization, M.T.; writing—original draft, M.T., R.E. and A.R.H.B.; writing—review & editing, R.E., A.R.H.B. and M.K.T. All authors have read and agreed to the published version of the manuscript.

**Funding:** This research and APC was funded by Kristiania University College, Oslo, Norway grant number 10093 ForskM 2022- SoMeSelf-408T.

**Institutional Review Board Statement:** The study was conducted by the Declaration of Helsinki and approved by the Norwegian Centre for Research Data (protocol code 644850 and date of approval 19th of May 2021) for studies involving humans.

**Informed Consent Statement:** Written informed consent was obtained from all participants involved in the study.

**Data Availability Statement:** The data presented in this study are available on request from the corresponding author.

**Acknowledgments:** The authors would like to acknowledge all participants for sharing their experiences regarding safe social media use.

**Conflicts of Interest:** The authors have no conflict of interest.

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
