# Peer review of "“Just Be Careful, Since Social Media Is Really Not as Safe as It’s Being Portrayed”: Adolescent Views on Adult Support for Safer Social Media Use"

_socsci, doi:10.3390/socsci11100492_

Round 1
Reviewer 1 Report
Dear authors,
I appreciated reading this paper, which I found well written and methodologically sound. I would just improve two parts:
1) Try to expand more the literature review. Right now you jump from the introduction to the methodology;
2) I would further stress the original contribution of the paper: are we actually learning something new? This needs to be clear.
Apart from that I think this is a good paper and wish the authors all the best!
Author Response
Dear Reviewer,
We would like to express our sincere gratitude for the valuable comments given on our manuscript. According to the suggestions from the reviewers, changes have been made and they are marked with tracking changes in the manuscript and in yellow colour.
1) Try to expand more the literature review. Right now you jump from the introduction to the methodology;
Authors' Response: We agree and we have added subheading 2. Literature review (lines 93-123). In this section, we have explained and described more relevant information from other researchers and rationale to support our aim with this study. We have added a few sentences in the introduction, too (lines 31-33 and 40-42).
2) I would further stress the original contribution of the paper: are we actually learning something new? This needs to be clear.
Authors' Response: Yes, we have learned something new. We have completed the conclusion (lines 469-481) and we know that the dissemination of knowledge about the safe use of social media use must become an integral part of adolescents’ education and part of the publicly available information in society.
Apart from that I think this is a good paper and wish the authors all the best!
Authors' Response: We are grateful for these nice words.
Reviewer 2 Report
Dear Author, I do not have any suggestions. However, it is very interesting why teens recognise their parents as outdated. After all, the parents were born after 1980, so they should be familiar with being online. How to explain it?

Author Response
We would like to express our sincere gratitude for the valuable comments given on our manuscript. According to the suggestions from the reviewers, changes have been made and they are marked with tracking changes in the manuscript and in yellow colour.
Dear Author, I do not have any suggestions. However, it is very interesting why teens recognise their parents as outdated. After all, the parents were born after 1980, so they should be familiar with being online. How to explain it?
Authors' Response: Thank you for your opinion. Unfortunately, we didn't investigate the age of next of kin. We are not sure if the parents were born after 1980. One of our explanations is that the rapidly evolving social media, especially the personalisation algorithms, is something too advanced and difficult to understand deeply. This can be one of the reasons.
The second of our explanations is that also somebody from us - researchers are born before 1980 and have children in teenager age. Our motivation was to ask other adolescents how they reflect on the safe use of social media, and we could "listen about ourselves." This study was incredibly interesting to understand how adolescents think about "our" abilities to help and support the safe use of social media. “We are in the same boat” (we’re all struggling with this problem) and we are learning by doing the same way as adolescents do.
We are very grateful for the positive feedback for our study.